# Detecting anomalous sea-level states in North Sea tide gauge data using an autoassociative Neural Network

Kathrin Wahle[1], Emil V. Stanev[1,2,3], Joanna Staneva[1]

[1]Helmholtz Zentrum Hereon, Geesthacht, Germany
[2] Research Department, University of Sofia "St. Kliment Ohridski", Sofia, Bulgaria,
[3] Department of Meteorology and Geophysics, University of Sofia "St. Kliment Ohridski", Sofia, Bulgaria

*Correspondence to*: Kathrin Wahle (kathrin.wahle@hereon.de)

**Abstract.**

The sea level in the North Sea is densely monitored by tide gauges. The data they provide can be used to solve different scientific and practical problems, including the validation of numerical models, and the detection of extreme events. This study focuses on the detection of sea-level states with anomalous spatial correlations using auto associative neural networks (AANNs), trained with different sets of observation- and model-based data. Such sea-level configurations are related to nonlinear ocean dynamics; therefore, neural networks appear to be the right candidate for their identification. The proposed network can be used to accurately detect such anomalies and localize them. We demonstrate that the atmospheric conditions under which anomalous sea-level states occur are characterized by high wind tendencies and pressure anomalies. The results show the potential of AANNs in accurately detecting the occurrence of such events. We show that the method works with AANN trained on tide gauge records as well as with AANN trained with model-based sea surface height outputs. The latter can be used to enhance the representation of anomalous sea-level events in ocean models. Quantitative analysis of such states may help assess and improve numerical model quality in the future as well as provide new insights into the nonlinear processes involved. The method has the advantage of being easily applicable to any tide gauge array without preprocessing the data or acquiring any additional information.

## 1 Introduction

The dynamics of sea level in tidal basins are one of the most addressed topics in physical oceanography. Theoretical prediction of tidal motion was pioneered by the application of Fourier analysis by Lord Kelvin (Thomson, 1880) and later improved by Doodsen (1921), who developed the tide-generating potential in harmonic form. Analysis and interpretation of tidal observations by Proudman and Doodson (1924) enhanced the understanding of sea-level fluctuations due to winds and changes in atmospheric pressure. The development of numerical 2D storm surge models by Peeck et al. (1983) and Flather and Proctor (1983) led to early warning systems for coastal flooding. With increasing computational power and the availability of satellite data, sea-level predictions have been continuously improved. However, current model predictions are not always perfect (Stanev et al., 2015; Sandery and Sakov, 2017; Staneva et al., 2016; Ponte et al. 2019; Mey et al., 2019; Jacobs et al., 2021), which emphasizes the need to further understanding of sea level.

A recent important evolution in predicting sea-level in the North Sea was achieved in the framework of the development of the Northwest European Shelf forecasting system (e.g. O'Dea et al., 2012, Tonani et al., 2019) by enhancing the model resolution to 1.5 km. Thus, dynamical features such as coastal currents, fronts, and mesoscale eddies are better resolved, and the model results are improved, especially when compared to high spatial–temporal resolution observations.

Satellite altimetry has added critical information in the last 30 years (Madsen et al., 2015). Notably, different measurement techniques have different advantages and disadvantages. Satellite-derived sea-level information, which has revolutionized oceanography and climate science, particularly in addressing global and large-scale change, is of limited use when addressing

near-coastal short-periodic variability. However, advancements are underway, and new satellite missions characterized by better spatial and temporal sampling have paved the way for improvements in coastal sea-level research (e.g. Dieng et al., 2021; Prandi et al., 2021; Dodet et al., 2020; Sanchez-Arcilla et al., 2021a).

Tide gauge stations operating along the North Sea coast provide high-quality records of sea-level observations over a long period (Wahl et al., 2013). Ponte et al. (2019), reviewing the state of science of coastal sea-level monitoring and prediction,

outlined the importance of sea-level observations for studying sea-level variability. However, tidal gauges do not provide information about the basin-wide patterns of sea level. Furthermore, some of these data are not continuous; different gauges do not always operate simultaneously, and there are gaps in many of the records. Therefore, a combination of numerical model results with tide gauge measurements would be beneficial. A similar exercise was undertaken recently by Madsen et al. (2019) for the Baltic Sea and by Zhang et al. (2020) for the North Sea. Zhang et al. (2020) used machine learning to reconstruct the

sea-level variability in the North Sea using observations from 19 coastal tide gauges and data from numerical models. Notably, they concluded that a relatively short-time record contains the most representative characteristics of sea-level dynamics in the North Sea. While this was clear for tides, it was not so obvious for changes in sea level caused by the atmosphere.

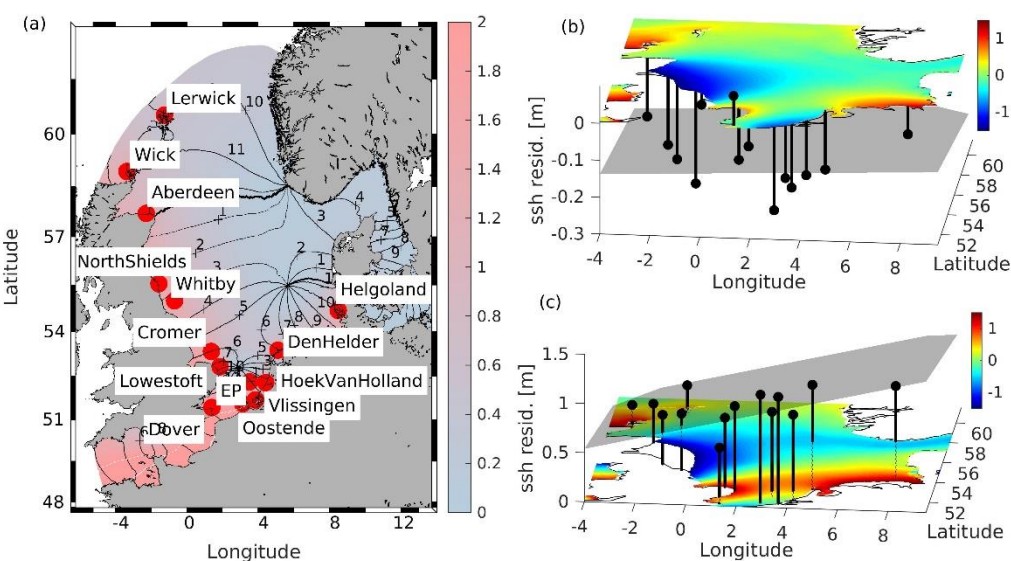

**Figure 1: (a) Location of tide gauge stations used in this study and M2 cotidal chart (M2 amplitudes (color scale) and phase lines (black isolines) reproduced from Jacob and Stanev, 2017). Additionally, snapshots of NEMO model water levels for two selected times are shown together with water level residuals at tide gauge locations (black lines) and a plane fitted linearly to these residuals (grey areas): (b) 7. January 2017, 16 UTC and (c) 11. January 2017, 20 UTC.**

The North Sea (Figure 1a) is a shallow sea with an average depth of ~ 90 m located at the European continental shelf. The dynamics of sea level in the North Sea, which is the region of our study, can be considered a coupled response to different forcings, such as barotropic tides, wind and atmospheric pressure, as well as forcings from the open boundaries and rivers, including a thermohaline forcing. The coupling of the respective processes is, in most cases, nonlinear (Jacob et al., 2017), therefore, one cannot easily identify the response to individual drivers in isolation. This happens when either oscillatory

motions have large amplitudes, e.g., tidal currents approaching 1 m/s, or when wind-driven current is of the same order. Thus, there is a need to use methods tailored to detect and reproduce nonlinear dynamics. Nonlinear processes are difficult to predict, even with sophisticated models; therefore, one needs to identify situations in which predictions fail (Ponte et al., 2019). Furthermore, it is interesting to understand the reasons for the deviation in the model predictions from the observations. The detection of such situations in data from a network of tide gauges is the aim of the present study.

The spatial and temporal correlations of tide gauge data mirror the complex sea-level dynamics in the North Sea (Figure 2). First, tidal wave propagation is influenced by the topography of the North Sea. The north-south decrease in water depth shifts

the central amphidromic point southwards (see Figure 1a for the oscillation pattern of the semidiurnal M2 tide). Two additional amphidromies are generated as a result of the superposition of incident and reflected Kelvin waves and cross-shore Poincare waves with centres in the north, off the Norwegian coast and in the south, between Suffolk and Holland (Figure 1a). The amplitude of the tidal wave increases towards the British coast. Off the Danish and Norwegian coasts, the amplitude of the wave is significantly smaller due to dissipation by bottom friction in the shallow southern part of the North Sea (Figure 2a). The circulation of the North Sea and thus the sea level is forced also by the atmosphere through either wind stresses (Figure 2c) or atmospheric pressure gradients (Figure 2d). Considering the length scales and depth of the North Sea, the wind has a dominant effect. The prevailing westerly winds result in a dominant cyclonic circulation. A reversal of circulation caused by easterly winds seldom occurs (Stanev et al., 2019). For northwesterly and southeasterly winds, circulation may stagnate. As shown by Jacob et al. (2017), the interaction between tidal and wind-driven currents is nonlinear in areas of strong currents, particularly in the German Bight, and the atmospheric variability affects the spring-neap variability more strongly than the M2 tide. Additionally, shallow-water tides show significant small-scale spatial variability patterns in this region (Stanev et al. 2015). Thus, in the southern North Sea, tidal forcing strongly impacts the residual circulation (Figure 2b). In summary, the specific topography of the North Sea together with nonlinear effects lead to complex circulation patterns that hamper sea-level prediction.

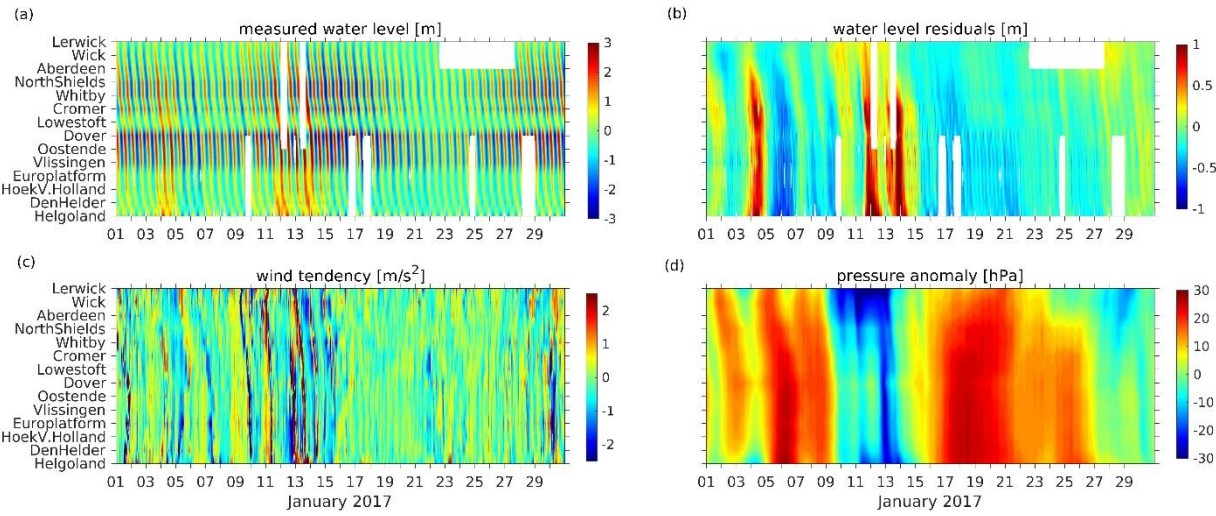

**Figure 2: Time versus position diagrams series of (a) measured water levels, (b) water level residuals, (c) wind tendency (defined as change in wind speed within one hour) and (d) pressure anomaly for January 2017 at tide gauge positions.**

Simple linear statistical methods, such as principal component analysis (PCA), fail to predict nonlinear dynamics because they represent linear combinations of mean states. Flinchem and Jay (2000) demonstrated that wavelet transforms provide an approach that is well suited for tidal phenomena that "deviate markedly from an assumed statistical stationarity or exact periodicity inherent in traditional tidal methods" and that it can also reveal features that harmonic analysis could not elucidate. In the same context, neural networks (NNs) have been used for (extreme) tidal surge prediction (French 2017, Tayel 2015, and Bruneau 2020). Similarly, Hieronymus et al. (2019) applied different machine learning techniques to the regression problem of time series data from tide gauges, and Balogun and Adebisi (2021) investigated the impact of different ocean-atmosphere interactions on sea-level predictability using different NNs. However, all these applications focus on data from single tide gauge stations.

Horsburgh (2007) proved the importance of spatial and temporal correlations among tide gauge stations along the British coast for the distribution of surge residuals. However, their linear model is only capable of describing some of the possible interactions between tides and surges and was used to demonstrate the existence of critical spatiotemporal scales for surge development and decay. Wenzel (2010) used neural networks for the reconstruction of monthly regional mean sea-level

anomalies from 59 tide gauges worldwide. Zhang et al. (2020) used generative adversal networks to reconstruct the sea level variability in the North Sea using observations from 19 coastal tide gauges and data from numerical models. Notably, they concluded that a relatively short-term record contains the most representative characteristics of sea-level dynamics in the North Sea. While this was clear for tides, it was not so obvious for changes in sea level caused by the atmosphere. Similarly, deep learning approach to fuse altimeter data with tide gauge data in the Mediterranean Sea was descrribed by Yang et al. (2021),

while Nieves (2021) et al. used open ocean temperatures to predict coastal sea-level variability via machine learning.

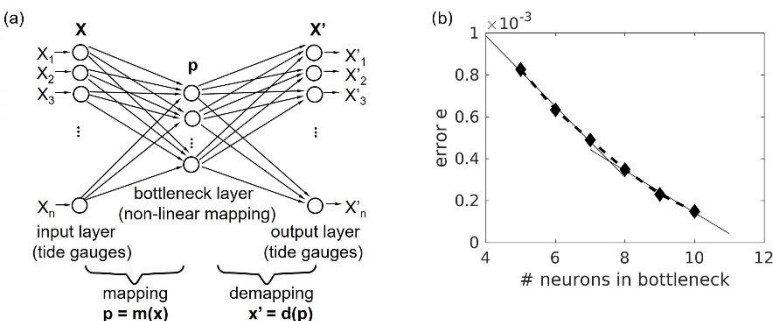

**Figure 3: (a) Autoassociative NN with bottleneck: the input is mapped by p = m(x) onto a lower dimensional space (dim(p)<dim(x))**
**and is approximately reconstructed by the demapping NN part x' = d(p) ≈ x. (b) Error measure from the training sample versus the**
**number of neurons in the AANN 'bottleneck'–layer using the 2016/17 North Sea water level data from the fourteen tide gauge**
**stations (see Fig.1).**

In contrast to these applications, we will focus on the identification of situations in which spatial correlations between tide gauge measurements deviate greatly from the principal correlations. Usually, tide gauge measurements in the North Sea show specific spatial correlations (changing in time with the tide). However, in anomalous situations (e.g., localized storms), these

correlations may drastically change. We use autoassociative neural networks (AANNs, Kramer, 1992) to detect such sea-level states. Similar to PCA, the aim of this method is dimensionality reduction; however, the concept of orthogonal vectors is expanded to principal curves. The combination of nonlinear components best describing the variability in the training data can be used to reconstruct the data. Thus, AANN provides a nonlinear reconstruction model, and the reconstruction error is a measure of how well the data are characterized. Large errors in the reconstruction of data by the AANN represents situations

in which the data do not belong to the same distribution they were built from. Atmospheric conditions related to such situations might aid further understanding and future developments in sea-level prediction.

The paper is structured as follows: In section 2, we present observational and model data used throughout our study and introduce the concept of AANN. We then apply the AANN to the tide gauge array measurements in the North Sea. In section

3, we use AANN to detect anomalous events and examine the dependence of identification of such events from the AANN training data used. Two events are studied in detail, including their relationship with the atmospheric conditions. This is followed by a discussion and conclusions.

## 2. Data and Methods

### 2.1 Observational and Model Data

Observational sea-level data along the North Sea coast have been obtained from the historical and near real-time (NRT) dataset of the Copernicus Marine Environment Monitoring Services (CMEMS). The observations were taken from the respective in situ products for the Northwest Shelf area (Copernicus Marine In Situ TAC Data Management Team, 2020, http://marine.copernicus.eu/) with hourly resolution. The NRT *in situ* quality-controlled observations are updated hourly and

distributed within approximately 24–48 h after acquisition. From these, we have chosen 14 gauge stations according to completeness of data time series. Their positions are shown in Figure 1a. Time versus position diagrams of measured and

detided (residual) water levels are shown in Figure 2a, b. Figure 1a shows the propagation of the tidal wave along the English coast with water level amplitudes increasing southwards towards the English Channel. The slope of contours in Figure 2 gives the speed of propagation of the tidal wave. The specific feature between the Cromer and Vilssingen stations identifies the small amphidrome in front of the English Channel. This feature is not present in the detided data (Figure 2b); the latter resembles the atmospheric forcing (Figure 2c, d). In the presence of large gradients in wind tendency (Figure 2c), the water surface tilts considerably (Figure 1c).

For the experiments discussed later in this study, we also use data from the GCOAST (Geesthacht Coupled cOAstal model SysTem) circulation, wave and ocean model (Madec, 2017, Staneva et al., 2017, 2021, Bonaduce et al., 2020). Wave-current interaction processes included in the model are momentum and energy sea state-dependent fluxes, wave induced mixing and Stokes-Coriolis forcing. The model area covers the Baltic Sea, the Danish Straits, the North Sea and part of the northeast Atlantic Ocean) with 3.5 km horizontal resolution. The data used in the present study are only for the North Sea region. The ocean circulation model is based on the Nucleus for European Modelling of the Ocean (NEMO v3.6). The wave model is WAM (cycle 4.7), a third-generation wave model, that solves the action balance equation without any *a priori* restriction on the evolution of the spectrum. The two models are two-way coupled via the OASIS3-MCT version 2.0 coupler (Valcke et al., 2013). The NEMO setup used within the GCOAST system uses an explicit free-surface formulation; 50 hybrid s-z* levels with partial cells are used in the vertical. Atmospheric pressure and tidal potential are also included in the model forcings (Egbert and Erofeeva, 2002). Daily river run-off is based on river discharge datasets from the German Federal Maritime and Hydrographic Agency (Bundesamt für Seeschifffahrt und Hydrographie, BSH), Swedish Meteorological and Hydrological Institute (SMHI) and United Kingdom Meteorological Office (Met Office). Boundary conditions at the open boundaries (temperature, salinity, velocity and sea level) are taken from the AMM7 model (O'Dea et al., 2012) distributed by the Copernicus Marine Environment and Monitoring Service. They have a temporal resolution of one hour with a 7 km horizontal resolution. The model forcings for momentum, and heat fluxes are computed using bulk aerodynamic formulae and hourly data from atmospheric reanalyses of the European Centre for Medium Range Weather Forecasts (ERA5, Hersbach et al, 2020), the 5th generation of reanalysis of the ECMWF. ERA5 has horizontal resolutions of 28 km and 31 km. We use ERA5 1-hourly products provided by Copernicus Climate Center Service (C3S, 2021) available with 0.25° and 0.25° horizontal resolutions for the atmospheric and wave parameters, respectively. Several studies have demonstrated the advantages of using the ERA5 reanalysis over its predecessor (ERA-INTERIM, e.g. Belmonte Rivas and Stoffelen, 2019).

**2.2 Autoassociative Neural Network**

Most environmental monitoring programs produce large sets of data, with the tide gauge data network in the North Sea being among them (CMEMS global ocean in situ near-real-time observations, https://doi.org/10.48670/moi-00036). Although these sea level data depend on a number of variables (tidal and atmospheric forcings, together with bathymetry, river input, etc.), this number is smaller than the number **n** of tide gauges within the network. The observed data are located within a subspace of the **n**-dimensional data space and can thus be compressed. Different machine learning techniques, such as the k-nearest neighbours algorithm, ensemble-based methods, and support vector machine (SVM) algorithms, are used to predict the posterior probabilities of a given dataset and are optimal for data compression. The different techniques are not equally well suited for detecting outliers, i.e., identifying whether a given observation belongs to the same probability distribution. Outlier detection in high dimensions, or without any assumptions about the distribution function of the data is very challenging. SVM algorithms work well if training data are not contaminated by outliers. Ensemble and k-nearest neighbour methods perform well for multimodal datasets. Covariance estimators, in which category PCA also falls, degrade when the data are not unimodal. Autoassociative neural networks (AANNs) combine the robust performance of multimodal data with the geometric interpretability of PCA identify these situations.

An AANN is a reconstruction model based on a feed-forward neural network (Kramer 1992). In an AANN, the **n**-dimensional

input **x** is mapped onto itself (**x'**) with a data compression step in between: The number of neurons in at least one hidden layer in an AANN is less than the dimension **n** of the input and output vectors **x** and **x'**. This layer is called the bottleneck-layer and is the key component of an AANN (Figure 3a). It provides data compression of the input with powerful feature extraction capabilities. The mapping part m(**x**) compresses the information of the **n**-dimensional vector to a smaller dimension subspace vector **p**, whereas the demapping part **x'**=d(**p**) uses compressed information to regenerate the original **n**-dimensional vector.


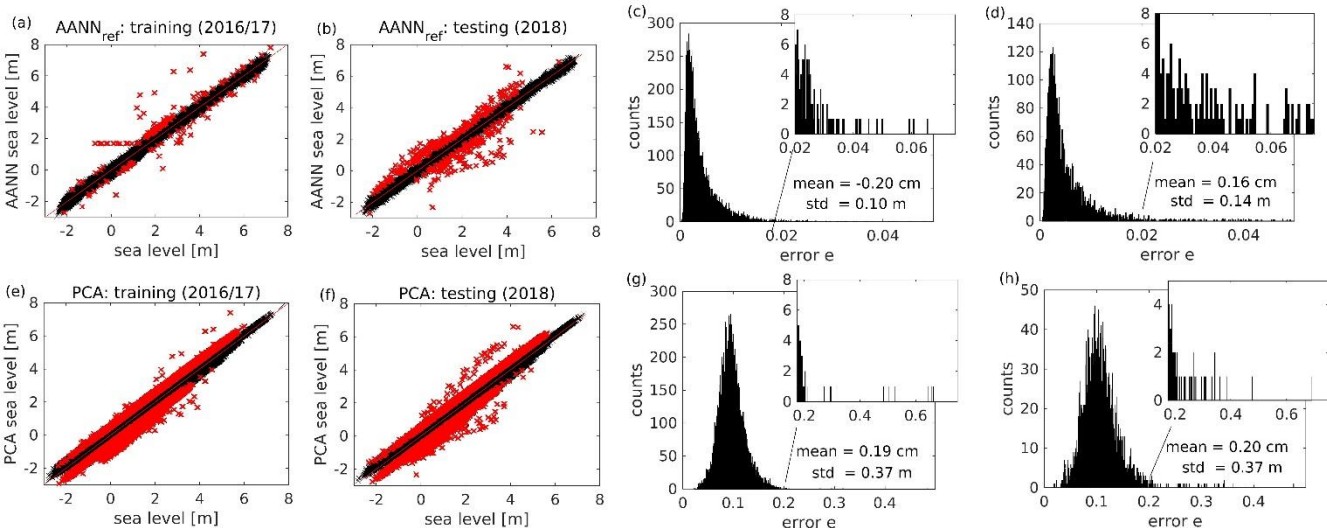

Figure 4: Performance of the reference AANN (a-d) and of the PCA (e-h) in reconstructing sea-level data during training (a, c, e, g) and testing (b, d, f, h) period. Red points indicate matchups with an absolute reconstruction error > 50.4 cm. This limit was derived from the 0.01% of training data with highest reconstruction error for the reference AANN. Histograms (c, d, g, h) show the respective
distributions of squared relative reconstruction errors e. The numbers are the mean error and standard deviation of the reconstruction. Insets show the tail of the distributions, containing possible candidates for anomalous sea states. Additionally, absolute mean errors and standard deviations are given.

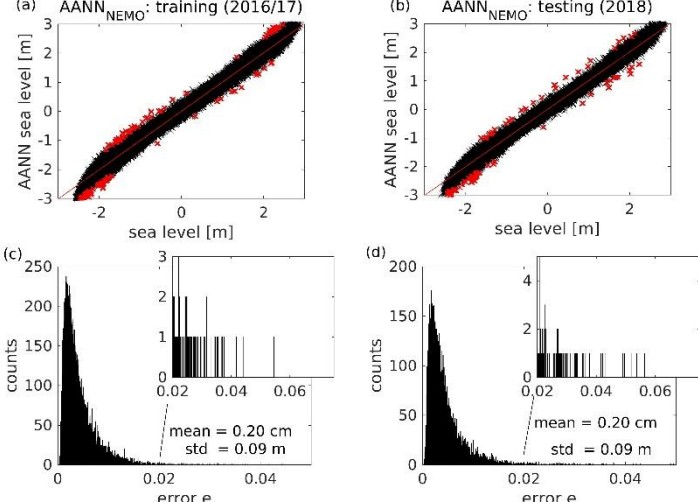

Figure 5: Same as Fig. 4 but for AANN_NEMO trained on NEMO modelled sea level data.

In the mapping part, the (nonlinear) correlations/functional relations existing among the input variables (here, the spatial correlations) are taken advantage of. Thus, an AANN can be considered to perform a nonlinear generalization of the PCA (nonlinear principal component analysis, NLPCA (Kramer 1991)). Similar to PCA, NLPCA can also serve important purposes, e.g., filtering noisy data, feature extraction, data compression, outliers and novelty detection (Mori 2016).

During training, the AANN constructs a model, that captures the posterior probability distribution of a given dataset.

At the beginning of the training, the outcome of the NN will differ largely from the target output. The mean squared relative error per neuron **e** is iteratively minimized during the training by backpropagating the error through the NN and adjusting free parameters according to a gradient descent scheme:

$$e = \frac{1}{N}\sum_{tr=1}^{N} e_{tr}\,, \quad \boldsymbol{e_{tr}} = \frac{1}{n}\sum_{i=1}^{n}\left(\frac{x_i' - x_i}{\max(x_i^{tr}) - \min(x_i^{tr}), tr \in [1,N]}\right)^2,$$

where N is the number of (training) input/output data pairs.

An important step in the construction of an appropriate AANN is the choice of the number of neurons in the bottleneck layer, i.e., to determine the dimension of the subspace **p**. The choice is based on the reconstruction abilities of the network: The overall reconstruction error of the AANN decreases with an increasing number of neurons in the bottleneck layer. However, if the number of bottleneck neurons equals the intrinsic dimensionality, a further increase leads to 'overfitting' of the AANN, i.e., learning stochastic variations in the dataset rather than the underlying functions. As a result, the decrease in the overall 215 error flattens (Figure 3b).

## 2.3 Application of AANN to the North Sea Water Level from Tide Gauges

The proposed scheme was applied to data from 14 (dim(**n**)=14) tide gauges along the North Sea coast (Section 2.1). Data from 2016/17 were used to train the AANNs, data from 2018 were retained as an independent testing set. A two-year 220 training period was assumed to be long enough to capture the essential spatial correlations among observations at different gauges well. Thus, any dominant factor affecting the interrelationship between observations from individual stations (bathymetry, the distance between stations, dominant winds and atmospheric pressure, etc.) should be reflected by the AANN. The choice of tide gauge locations was a compromise between spatial coverage and data availability (since missing data from single gauge stations were not filled, e.g., by interpolation, and thus resulted in data loss). We applied no further 225 preprocessing or quality control to the tide gauge data.

A sequence of AANNs with an increasing number of neurons in the bottleneck layer was trained to map the 14 water levels onto themselves. In Figure 3b, the relative reconstruction error **e** for all training data is plotted versus the number of neurons in the bottleneck layer. Error decreases with an increasing number of bottleneck neurons; the decrease starts to flatten at seven neurons (dim(**p**)=7) in the bottleneck layer. Therefore, we decided to use seven bottleneck neurons and refer to this as 230 AANN_ref (reference AANN).

## 2.4 Quality of AANN Analysis

The performance of the AANN_ref is shown in Figure 4. The scatterplots are used to compare the modelled vs. observed sea-level data. The distribution for the training phase (Figure 4a) closely follows the bisectrix and is very narrow. The standard 235 deviation between the observed and AANN reconstructed sea-level data is approximately 10 cm (Figure 4c). For the testing phase (Figure 4b, d), the agreement between the observations and modelled data is almost as good as that for the training phase; however, some scatter is also clearly seen. Parts of this disagreement, with errors above the 99.9 percentile during the training phase, are marked in red and indicate possible candidates for anomalous situations, which will be further analysed. The distribution of errors is non-Gaussian during training and testing (Fig. 4c,d). The long tail of the error histogram shows 240 data where the reconstruction by AANN failed and where anomalous correlations among tide gauges are expected.

The non-Gaussian nature of the error distribution demonstrates the capability of the AANN to capture complex (usually nonlinear) processes. This is the fundamental difference from the case when PCA (using 7 PCs, analogous to the number of bottleneck neurons in AANN_ref) is used to reconstruct the observed data (Figure 4e-h). In PCA, the results have approximately 3.5 times higher standard deviations. PCA here leads to a short-tailed Gaussian error distribution (Figure 4g, 245 h), making it unsuitable for the desired task. Because the non-Gaussian distribution is a typical characteristic of nonlinear

processes, and because linear methods (PCA) do not capture it, we conclude that the above comparison gives a clear demonstration of the power of the AANN method used in the present study to handle nonlinearities. Outliers are usually in the tail of the distribution, and their identification requires using nonlinear reconstruction methods.

## 3. Anomalous Sea-Levels and Their Relationship with Atmospheric Conditions

To isolate events that strongly affect the spatial correlations among tide gauges, further constraints are needed. High values of AANN_ref reconstruction errors might originate from events on small scales, such as ocean response to wind gusts, as well as the erroneous measurement of a single tide gauge. From the analysis of the training period, we postulate an ocean state as anomalous when the **error e exceed 0.035 (see Figures 4c, d) for at least 3 gauges simultaneously for at least 3 hours**.

To exclude the dependence of identification of such extreme events from the data used, we trained alternative AANNs based on the following datasets:

- trained with data extracted at the closest positions of gauge stations from the CMEMS operational model AMM15 (AANN_NEMO);
- trained with a a subset of gauge data from only 10 stations (with Dover, Oostende, Vlissingen and Europlatform in the southwest corner of the North Sea excluded) (AANN_less); and

trained with detided data from the 14 gauge stations (AANN_resid). Software package T-TIDE (Pawlowicz et al., 2002) was used for this purpose.

As for AANN_ref, training data span the two-year period 2016/17 and data from 2018 were kept for validation.

The first additional network AANN_NEMO allows an intercomparison between observations and simulations. If the agreement is good, spatial data from the operational model will enable the analysis of horizontal patterns associated with the identified events. The second network, AANN_less, identifies the possible occurrence of anomalous events in the southwestern North Sea area. This area is known to be dominated by a small amphidrome (Figure 1a) and is characterized by its complex dynamics and high amplitudes of tidal oscillations. The third network, AANN_resid, reveals how strongly anomalous events are associated with the basic tidal dynamics, which are removed here.

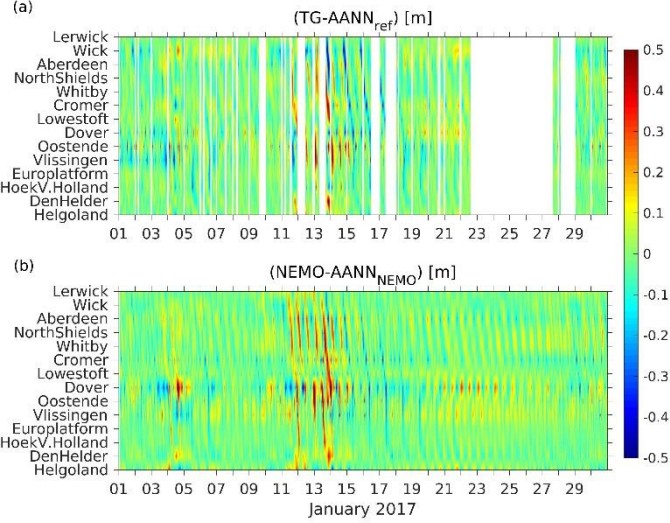

**Figure 6: Time versus position diagrams of differences between (a) measured water levels and AANN_ref emulated ones and between (b) NEMO modelled and AANN_NEMO emulated ones for January 2017.**

The error distributions of AANN_less, AANN_resid and AANN_NEMO are similar to those of AANN_ref. The tail of the AANN_NEMO error distribution is shortened, making modelling data less suitable for novelty detection (see Figure 5c,d). A comparison of AANN_NEMO and AANN_ref errors is shown in Figure 6. Anomalous situations detected with AANN_ref (4., 11.-14. January 2017) reflect themselves in large (exceeding 25 cm) AANN_NEMO errors but not necessarily vice versa.

**Table 1: Overview of the number of anomalous sea states detected by the alternative training dataset AANN and the reference model AANN_ref approaches during the 2018 testing phase.**

|  | training dataset | Spring (MAM) | Summer (JJA) | Autumn (SON) | Winter (DJF) |
|---|---|---|---|---|---|
| AANN_NEMO | NEMO model (14 data points) | 2 | 2 | 6 | 13 |
| AANN_less | tide gauge data (raw, 10 gauges) | 1 | 0 | 6 | 13 |
| AANN_resid | tide gauge data (de-tided, 14 gauges) | 1 | 1 | 6 | 13 |
| **AANN_ref** | **tide gauge data (raw, 14 gauges)** | **2** | **2** | **6** | **13** |

The proposed criterion was tested on the reference model AANN_ref for the 2018 testing period. Several anomalous sea states were detected (Table 1). Events in the winter months occur more than twice as often as in other seasons. Afterwards, it was verified whether a specific event could be observed with AANNs trained on alternative datasets (see Table 1). Due to the short-tailed error distribution, all events can be detected with AANN_NEMO (but as a stand-alone method, it gives a larger number of false alarms). Events from autumn and winter are also detected with the alternative AANNs (AANN_resid and AANN_less). This does not apply to spring and summer, where some events are not detected with AANN_less and/or by AANN_resid.

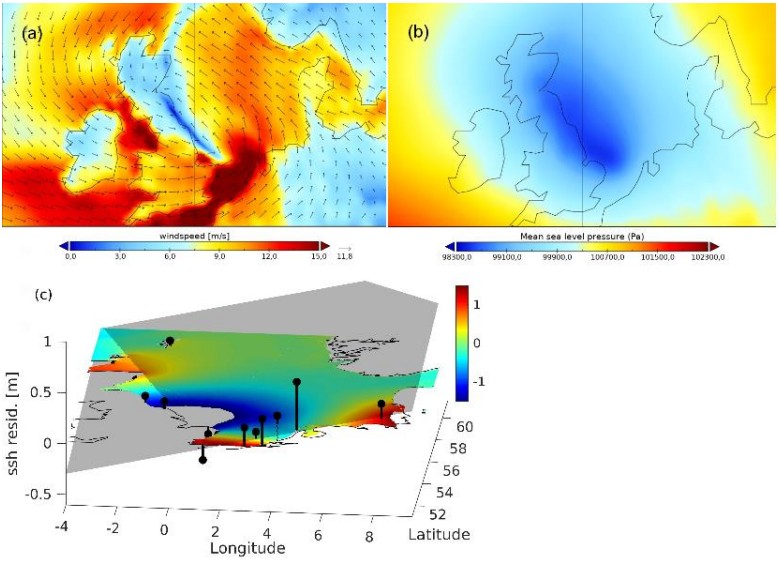

**Figure 7: Snapshots show the wind (a) and pressure (b) fields at the time of the largest wind tendency occurrence (6. June 2017, 12 UTC). Relatedly, the snapshot (c) shows NEMO ssh at the time of largest AANN error (6. June 2017, 22 UTC) together with tide gauge residuals (black lines). The grey area shows the orientation of a plane fitted linearly to these residuals.**

In the following, we will analyse two such events: The first event from June 2017 (low pressure 'Heinrich') was chosen since it was detected by AANN_resid but not by AANN_less (Figure 7 and, 8c, d). This atmospheric system brought some unseasonably wet and windy weather from France to the British southeast coast through the Channel. The rain was accompanied by very strong westerly winds (~15 m/s), with gusts of 20–25 m/s around the coast of England and Wales (see Figure 7a, b and Figure 8c,d). As a result, the measured sea-level (Figure 8a) exceeded the AANN_ref modelled sea-level over several hours along the western Dutch coast and southeastern British coast (Figure 9a). The AANN_ref and AANN_NEMO (Figure 9b) errors are largest at Oostende, where the measured low tide is higher than expected by the AANNs. The occurrence of the largest error shifts to Dover for AANN_resid and to Den Helder for PCA (Figure 9e, f). The NEMO model error (Figure 9c) is the largest for Vlissingen, where it under- and overestimates the measured values for high and low tides, respectively. Consistently, AANN_less does not detect the event (Figure 9d), indicating its origin in anomalous spatial correlations among gauges around the small amphidrome. Thus, the specific forcing resulted in a localized (exceptional) increase in sea level at gauge station Den Helder. The snapshot in Figure 7c visualizes this finding at the time the AANN largest errors occurred, 10 hours after the wind tendency.

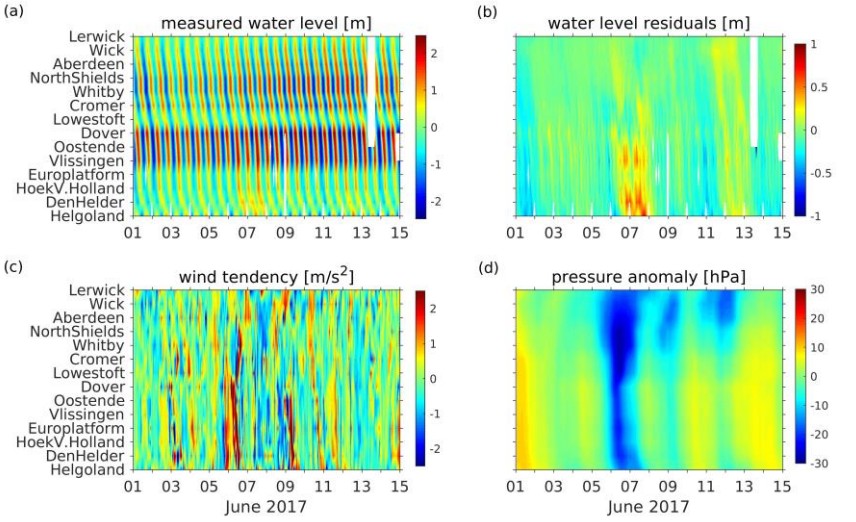

**Figure 8: Time versus position diagrams of measured tide gauge data (a), its residuals (b) and atmospheric forcing (ERA 5 data, wind tendency (c) and pressure anomaly (d)) for 1.-15. June 2017.**

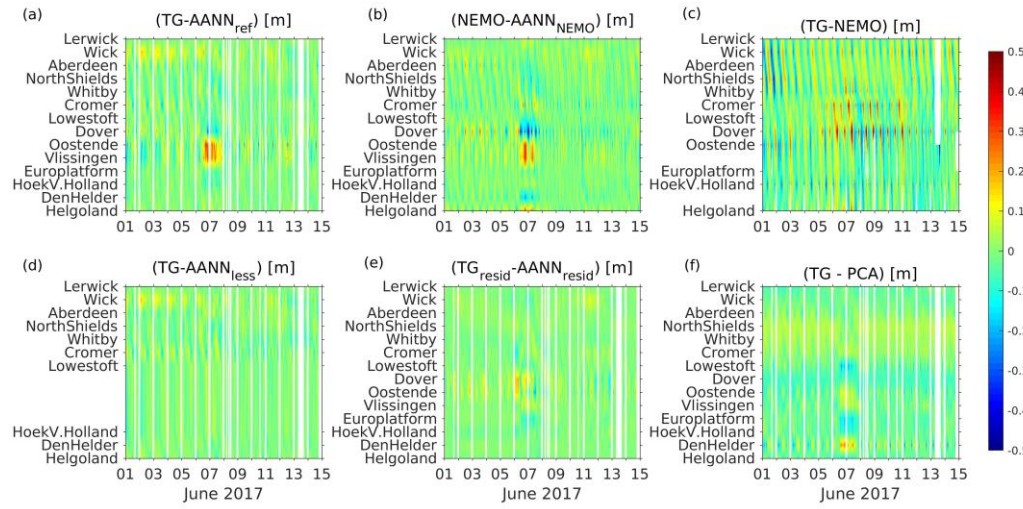

**Figure 9: Time versus positions diagrams of different reconstruction errors with respect to measured (detided) and NEMO modelled tide gauge data for 1.-15. June 2017: (a) AANN_ref, (b) AANN_nemo (trained on modelled data), (c) NEMO model, (d) AANN_less (without 4 stations close to the first amphidrome) (e) AANN_resid (trained on water level residuals), and (f) using PCA, respectively.**

As a second example, a typical winter storm was chosen: the deep depression 'Burglind' (Figures 10a, b and 11c, d), which

was associated with high wind speeds (up to 20 m/s) along the English and the Dutch coasts but also with large gradients in the wind field (Figure 7c) with gusts exceeding 33 m/s in England and Wales. The storm track passed over Ireland, crossed the United Kingdom and then moved over Central Europe and dissipated.

As a result, the measured tide gauge values exceeded the AANN_ref, AANN_NEMO and AANN_less values at stations along the southern English and Dutch coasts (Figure 12a, b, d). Exceptions are Lowestoft and Cromer in the 'shadow' of the cyclone.

The NEMO model results (Figure 12c) show good agreement during high tide, especially along the Scottish coast, but tend to overestimate the low tide values, especially along the Dutch coast.

AANN_resid and PCA (Figure 12e, f) also detects the event. However, here, stations along the Dutch coast have the largest reconstruction errors (i.e., stations with the highest water level residuals, see Figure 11b). The snapshot in Figure 10c visualizes water level residuals at the time of the largest AANN errors, which occurred with a 7 hour delay to wind tendency.


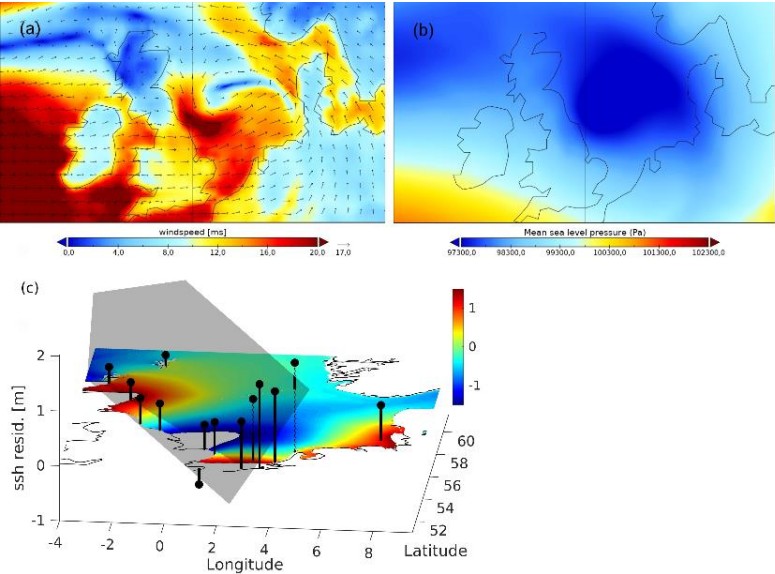

**Figure 10: Same as Figure 7, but at the time of the largest wind tendency occurrence (3. January 2018, 6 UTC) and largest AANN reconstruction error (3. January 2018, 13 UTC) during storm 'Burglind'.**

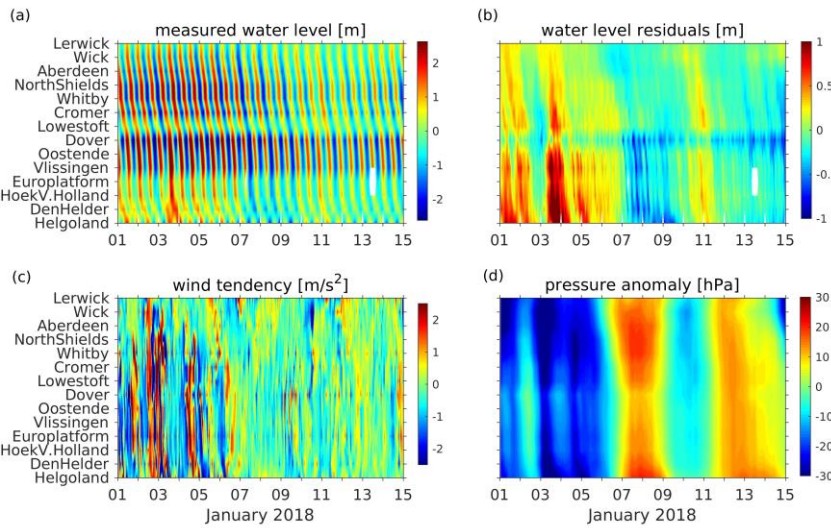

**Figure 11: Same as Figure 8, but for 1.-15. January 2018.**

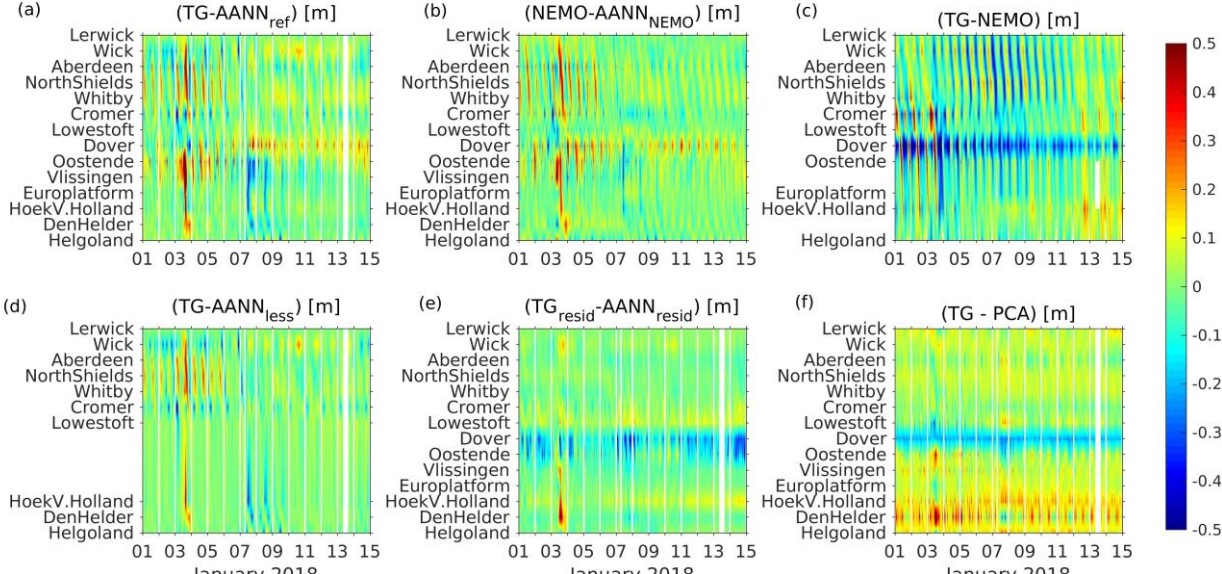

**Figure 12: Same as Figure 9, but for 1.-15. January 2018.**

On either occasion, a typical time delay of 5 to 15 hours between large tendencies in wind forcing and large reconstruction errors is observed, supporting the assumption that the atmosphere plays a key role in the detected events. However, their occurrence cannot be directly linked to atmospheric variables, such as large tendencies in the wind forcing. These tendencies occur often but seldom result in such anomalous correlations between tide gauge measurements.

**Conclusions**

We presented a tool for detecting anomalous water levels from a network of tide gauges in the North Sea using AANN. Strong spatial correlations between gauges allow for a reconstruction model in a lower dimensional subspace. Combining a threshold value for reconstruction error with the requirement of its occurrence at three different gauges for at least three hours has been shown to be a valuable filter for such events. For the two events discussed in detail, atmospheric conditions showed high wind

tendencies and pressure anomalies, with the accumulation of water masses in specific areas. Thus, the resultant sea states are assumed to be related to nonlinear interactions among the various atmospheric and oceanic forcings. Further analysis is needed to develop a deeper understanding of the underlying processes. Ultimately, this understanding might improve numerical sea-level predictions. The AANN reveals an intrinsic dimensionality of $\mathbf{p}$=7. Evident variable candidates that could be involved in further studies include location (latitude/longitude), time, wind tendency, pressure anomaly and information on water currents.


In addition, the tool offers an inexpensive opportunity to monitor the tidal gauge array. The difference between using raw and detided data as input to the model was marginal. Thus, no further data preprocessing is needed. The AANN model reacts sensitively to the choice of tide gauge locations: AANN_less, where stations around the first amphidrome were removed from the training data, was not able to identify events resulting from storms along the channel.


The linear PCA reconstruction method reveals a high rate of false positive alarms, which indicates a multimodal probability distribution of the sea-level data. For similar reasons, the detection of some of these events might fail for AANN_NEMO where the underlying data are consistent throughout space and time. On the other hand, this finding substantiates the hypothesis, that AANN is able to detect situations under which model physics need further improvement.

The proposed method can be adapted easily to any tide gauge array. However, the intrinsic dimensionality of the constructed AANN might differ, as well as the involved forcings and underlying processes.

**Competing interests.** The corresponding author has declared that neither they nor their coauthors have any competing interests.


**Disclaimer.** Publisher's note: Copernicus Publications remains neutral with regard to jurisdictional claims in published maps and 10 institutional affiliations.

**Acknowledgements.** We acknowledge the Cluster of Excellence EXC20 2037 "Climate, Climatic Change, and Society"
(CLICCS) (project no. 390683824). ES acknowledges the EU H2020 project DOORS (grant no. 101000518). JS acknowledges the European Green Deal project "Large scale RESToration of COASTal ecosystems through rivers to sea connectivity" (REST-COAST) (grant no. 101037097). KW gratefully acknowledges funding from the EU H2020 project IMMERSE (grant no. 821926). We would like to thank S. Grayek for providing NEMO model output data.

**Financial support:** This research has been supported by the European Green Deal project "Large scale RESToration of COASTal ecosystems through rivers to sea connectivity" (REST COAST) (grant no. 101037097) and the  EU H2020 project IMMERSE (grant no. 821926).

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
