# Peer review of "Detecting anomalous sea-level states in North Sea tide gauge data using an autoassociative Neural Network"

_EGUsphere, 2022_

## Referee Comment (RC1)

**Detecting anomalous sea-level states in North Sea tide gauge data using of auto associative Neural Network**

**Review**

The manuscript aims to investigate the skill of Auto Associative Neural Networks (AANNs), trained with different set of observation- and model-based data, to emulate sea-level and detect extremes in the North Sea. The method proposed by the authors relies on the capabilities of linear and non-linear models based on principal components (PCs) and AANNs, respectively, to reconstruct sea-level states. The departures between the reconstructed and observed sea states are then used to detect extreme events in the North Sea. The results, focused on two events characterized by low pressure systems and high wind speeds over the North Sea, show the potential of AANNs trained with tide-gauge records in detecting accurately the occurrence of extremes, both as a very localized and larger scale events, while underline the inaccuracies of the emulators based on PCs (false positives) and trained with model-based sea-surface height outputs due to model physics (Gaussian distribution of errors). The latter can be used to enhance representation of sea-level extreme events in ocean models. The authors highlight the high potential of their approach which can be easily extended to other sea-level observing networks.

The work is well developed, and the publication is recommended after a minor revision of the manuscript

Here follow some suggestions, which could help improve certain parts of the manuscript.

**General Comments**

1. A few sentences (e,g in the Introduction and Section 2) should be rephrased, and relevant references to the literature should be acknowledged (see Specific Comments).

2. Sometimes test, validation, and control are used as synonyms: check it and ensure consistency.

3. The trained AANNs used in this work should be named appropriately (e.g. reference) and summarized in a dedicated Table giving a short description of the data sets used in training and testing phase.

4. Description of error distribution of AANNs (Figure 4): those results guide your analysis and conclusions, but it is hard to distinguish between the tail of error distributions in Figure 3 and Figure 4. This is probably due to poor quality of the graphics (see specific comments about Figures), which should be improved, but a more extended description of the results would help the reader to capture the actual meaning of the results.

5. The organization of text in the different Sections should be revised (see Specific Comments).

**Specific Comments**

**Introduction**

**Line 17:** "tidal motion …. tidal motion" - rephrase.

**Line 22-23**: add relevant references.

**Line 25-26**: explain shortly the improvements you are mentioning.

**Line 31-34:** Rephrase (e.g. change the order of the sentences)

**Line 34:** the sentence on numerical models can move further down in the text

**Line 38-42:** Rephrase

**Line 42-43:** This sentence can move e.g. to line 37: "Ponte et al., (2019) …. Sea-level variability. Therefore,..."

**Line 44:** This sentence should be linked to the findings of previous work (e.g Zangh et al. 2020).

**Line 47-51:** Rephrase and add relevant references to the literature.

**Line 58-59:** Add references.

**Line 83-84:** Rephrase.

**Section 2.1**

**Line 111**: Rephrase.

**Section 2.2**

**Line 134:** add references.

**Line 143:** avoid "very"**.**

**Line 144:**  Rephrase**.**

**Line 144:** check punctuation.

**Line 155:** avoid repetitions: "Thus,…"

**Line 192:** "presents" → "shows"

**Line 204**: Is this choice based on a sensitivity analysis? The choice of the constraints/threshold errors should be explained in the text.

**Line 207-2011:** "…, we trained the AANN based on the following data sets:

- sea-surface height outputs from CMEMS AMM15 (operational model) extracted at the tide-gauge positions (AANN_NEMO);
- observed total water levels from only 10 tide-gauge stations (with …) (AANN_less); and
- observed non-tidal water level residuals for the 14 gauge stations (AANN_resid). "

**Line 2015:** "areal" ?

**Lines 225**: "…a comparison between observed, modelled and AANN emulated…".

**Line 232:** "the reference model" ?. I assume this refers to AANN in Table 1 (See General Comments).
**Line 239:** "In the following Section,…": in the current version of the manuscript this Section is followed by the Conclusion. This part of the text should be reorganized.

**Line 240:** It is unclear which ANN model error you are referring to (ANN_resid ?). The same applies tp NEMO model error: is it the error in NEMO ssh outputs?

**Comments about Figures**

**Figure 1:** The description in the caption should be improved as well as the labels in the figure (see wind tendency).

**Figure 3:** It is hard to read the labels.

**Figure 4:**

- Panels a, b, d, e (see previous comment on Figure 3).
- Panel c: It is very hard to read this panel, both time-series and labels. Also: show hours on x-axis.

**Figure 5:** The panels should be displayed bigger as well ass the labels in each of them.
Caption:
- "…a 14-day…"
- "NEMO model error" – is it the AANN_NEMO ? Clarify.

**Figure 6:** Make sure the labels on top of the panels correspond with the description in the caption.

**Figure 7:** see comments on Figure 5.

**Figure 8:**  see comments on Figure 6.

**Figure 9:** The panels can be displayed better. Avoid legend.

---

## Author Comment (AC1)

**Detecting anomalous sea-level states in North Sea tide gauge data using of auto associative Neural Network**

**Dear Agustín Sánchez-Arcilla, Dear Reviewer,**

**Thank you motivatiforng us to improve further our paper, for reading it so carefully and for your valuable comments. We have corrected the manuscript accordingly and provide the answers below.**

**Looking forward for any further instructions.**

**Best regards,**

**Kathrin Wahle**

**Review #1**

The manuscript aims to investigate the skill of Auto Associative Neural Networks (AANNs), trained with different set of observation- and model-based data, to emulate sea-level and detect extremes in the North Sea. The method proposed by the authors relies on the capabilities of linear and non-linear models based on principal components (PCs) and AANNs, respectively, to reconstruct sea-level states. The departures between the reconstructed and observed sea states are then used to detect extreme events in the North Sea. The results, focused on two events characterized by low pressure systems and high wind speeds over the North Sea, show the potential of AANNs trained with tide-gauge records in detecting accurately the occurrence of extremes, both as a very localized and larger scale events, while underline the inaccuracies of the emulators based on PCs (false positives) and trained with model-based sea-surface height outputs due to model physics (Gaussian distribution of errors). The latter can be used to enhance representation of sea-level extreme events in ocean models. The authors highlight the high potential of their approach which can be easily extended to other sea-level observing networks.

The work is well developed, and the publication is recommended after a minor revision of the manuscript

Here follow some suggestions, which could help improve certain parts of the manuscript.

**General Comments**

1. A few sentences (e,g in the Introduction and Section 2) should be rephrased, and relevant references to the literature should be acknowledged (see Specific Comments).
**We rephrased many sentences throughout the paper to improve its quality and readability and added relevant references.**

2. Sometimes test, validation, and control are used as synonyms: check it and ensure consistency.
**Thank you for the comment. For consistency, we used in the revised manuscript 'test'**

3. The trained AANNs used in this work should be named appropriately (e.g. reference) and summarized in a dedicated Table giving a short description of the data sets used in training and testing phase.
**We added to Table 1 the names of AANN's and their dataset. The training (2016/2017) and validation (2018) phases are the same for all NNs. We clarified and added this information, too:**

**"As for AANN_ref, training data in the latter three networks span the two year period 2016/17 and data from 2018 were kept for testing generalization abilities."**

4. Description of error distribution of AANNs (Figure 4): those results guide your analysis and conclusions, but it is hard to distinguish between the tail of error distributions in Figure 3 and Figure 4. This is probably due to poor quality of the graphics (see specific comments about Figures), which should be improved, but a more extended description of the results would help the reader to capture the actual meaning of the results.

**Thank you for the comments. We have replotted this (and all other) Figure(s) with higher resolution and better quality; we enlarged the insets and provided an extended Figure caption.**

5. The organization of text in the different Sections should be revised (see Specific Comments).

**Specific Comments**

**Introduction**

**General comment to Introduction:**
**Lines 27—44: We have restructured this part of the introduction, separating now numerical models from satellite measurements and tide gauges.**

Line 17: "tidal motion …. tidal motion" - rephrase.

**We rephrased as: ". Theoretical prediction of tidal motion was pioneered by the application of Fourier analysis by Lord Kelvin (Thomson, 1880) and later improved by Doodsen (1921), who developed the tide-generating potential in harmonic form."**

Line 22-23: add relevant references.

**We added the following recent relevant publications:**
**Sandery P.A., Sakov P. (2017)Ocean forecasting of mesoscale features can deteriorate by increasing model resolution towards the submesoscale. Nature Commun., 8, pp. 1-8**
**Stanev EV, F Ziemer, J Schulz-Stellenfleth J Seemann, J Staneva and KW Gurgel (2015) Blending surface currents from HF radar observations and numerical modelling: Tidal hindcasts and forecasts. Journal of Atmospheric and Oceanic Technology, Vol. 32, 256-281.**
**Mey-Frémaux, P. de, Ayoub, N., Barth, A., Brewin, R., Charria, G., Campuzano, F., Ciavatta, S., Cirano, M., Edwards, C.A., Federico, I., Gao, S., Hermosa, I.G., Sotillo, M.G., Hewitt, H., Hole, L.R., Holt, J., King, R., Kourafalou, V., Lu, Y., Mourre, B., Pascual, A., Staneva, J., Stanev, E.V., Wang, H., & Zhu, X. (2019): Model-Observations Synergy in the Coastal Ocean. Front. Mar. Sci., 23 July 2019, doi:10.3389/fmars.2019.00436**
**Jacobs, G., D'Addezio, J.M., Ngodock, H. and Souopgui, I. (2021). Observation and model resolution implications to ocean prediction. Ocean Modelling, 159, 101760, Ponte, R. M., Carson, M., Cirano, M., Domingues, C. M., Jevrejeva, S., Marcos, M., ... and Zhang, X.: Towards comprehensive observing and modeling systems for monitoring and predicting regional to coastal sea level. Frontiers in Marine Science, 6, 437, 2019.**

Line 25-26: explain shortly the improvements you are mentioning.

**We added, that by enhancing model resolution to 1.5 km, dynamical features such as coastal currents, fronts, and mesoscale eddies are better resolved, and model results improve, especially when compared to high spatial–temporal resolution observations.**

Line 31-34: Rephrase (e.g. change the order of the sentences)

**We rephrased: "However, advancements are underway, and new satellite missions characterized by better spatial and temporal sampling pave the way for improvements in coastal sea-level research (e.g. Dieng et al., 2021; Prandi et al., 2021; Dodet et al., 2020, Sanchez-Arcilla et al., 2021a, ).**

Tide gauge stations operating along the North Sea coast provide high-quality records of sea level observations over a long period (Wahl et al., 2013). Ponte et al. (2019), reviewing the state of science of coastal sea-level monitoring and prediction, outlined the importance of sea-level observations for studying sea-level variability. However, tidal gauges do not provide information about the basin-wide patterns of sea level. Furthermore, some of these data are not continuous; different gauges do not always operate simultaneously and there are gaps in many of the records.

**Line 34:** the sentence on numerical models can move further down in the text

We extended the paragraph on recent developments in numerical modelling (Lines 37ff) and added the information there: "Recent important evolution in predicting sea-level in the North Sea was achieved in the framework of the development of the Northwest European Shelf forecasting system (e.g. O'Dea et al., 2012, Tonani et al., 2019) by enhancing model resolution to 1.5 km. Thus dynamical features such as coastal currents, fronts, and mesoscale eddies are better resolved, and improve model results, especially when compared to high spatial–temporal resolution observations."

**Line 38-42:** Rephrase

We now state: "Therefore, a consistent dataset combining the gains of numerical model results with tide gauge measurements would be beneficial."

**Line 42-43:** This sentence can move e.g. to line 37: "Ponte et al., (2019) …. Sea-level variability. Therefore,…"

Yes. It now reads: "Tide gauge stations operating along the North Sea coast provide high-quality records of sea level observations over a long period (Wahl et al., 2013). Ponte et al. (2019), reviewing the state-of-the- of science of coastal sea-level monitoring and prediction, outlined the importance of sea-level observations for studying sea-level variability. However, tidal gauges do not provide information about the basin-wide patterns of sea level. Furthermore, some of these data are not continuous; different gauges do not always operate simultaneously; , and there are gaps in many of the records. Therefore,…"

**Line 44:** This sentence should be linked to the findings of previous work (e.g Zangh et al. 2020).

Yes. We link the two sentences, by:

"Zhang et al. (2020) use machine learning to reconstruct the sea-level variability in the North Sea using observations from 19 coastal tide gauges and data from numerical models. Noteworthy, they concluded that a relatively short-time record contains the most representative characteristics of sea level dynamics in the North Sea. While this was clear for the tides, it was not so obvious about changes in sea level caused by the atmosphere."

**Line 47-51:** Rephrase and add relevant references to the literature.

We rephrased:

"The coupling of the respective processes is, in most cases, nonlinear (Jacob et al., 2017), that is, one cannot easily consider the response to individual drivers in isolation. This happens, when either oscillatory motion has large amplitudes, e.g., tidal currents approaching 1 m/s, or wind driven current is of the same order. Thus, there is a need to use methods tailored to detect and reproduce nonlinear dynamics. The nonlinear processes are difficult to predict, even with sophisticated models; therefore, one also has to identify situations in which predictions fail (Ponte et al., 2019). Furthermore,…"

Jacobs, G., D'Addezio, J.M., Ngodock, H. and Souopgui, I. (2021). Observation and model resolution implications to ocean prediction. Ocean Modelling, 159, 101760,

Ponte, R.M., Carson, M., Cirano, M., Domingues, C.M., Jevrejeva, S., Marcos, M., Mitchum, G., Wal, R.S.W. van de, Woodworth, P.L., Ablain, M., Ardhuin, F., Ballu, V., Becker, M., Benveniste, J., Birol, F., Bradshaw, E., Cazenave, A., Mey-Frémaux, P. de, Durand, F., Ezer, T., Fu, L.-L., Fukumori, I., Gordon, K., Gravelle, M., Griffies, S.M., Han, W., Hibbert, A., Hughes, C.W., Idier, D., Kourafalou, V.H., Little, C.M., Matthews, A., Melet, A., Merrifield, M., Meyssignac, B., Minobe, S., Penduff, T.,

**Picot, N., Piecuch, C., Ray, R.D., Rickards, L., Santamaría-Gómez, A., Stammer, D., Staneva, J., Testut, L., Thompson, K., Thompson, P., Vignudelli, S., Williams, J., Williams, S.D.P., Wöppelmann, G., Zanna, L., & Zhang, X. (2019): Towards Comprehensive Observing and Modeling Systems for Monitoring and Predicting Regional to Coastal Sea Level. Front. Mar. Sci. 6:437, doi:10.3389/fmars.2019.00437**

**Line 58-59:** Add references.

**We reformulated this paragraph and moved it to the end of the introduction. We added reference on AANN:**

**"In contrast to these applications, we will focus on the identification of situations in which spatial correlations between tide gauge measurements deviate greatly from the dominant principal ones. Usually, the entirety of tide gauge measurements in the North Sea will show specific spatial correlations (changing in time with the tide). However, in anomalous situations (e.g. localized storms) these correlations may drastically change. We use autoassociative neural nets (AANNs, Kramer, 1992) to detect such sea-level states. Atmospheric conditions related to such situations might aid further understanding and future developments in sea-level prediction."**

**Kramer, M. A.: Autoassociative neural networks. Computers & chemical engineering, 16(4), 313-328, 1992.**

**Line 83-84:** Rephrase .

**We rephrased as :**

**"The paper is structured as follows: In section 2 we present the observational and model data used throughout our study and introduce the concept of AANN. We then apply AANN onto the tide gauge array measurements in the North Sea. In section 3 we use AANN to detect anomalous events and examine the dependence of identification of such events from the AANN training data used. Two events are studied in detail, including atmospheric conditions. This is followed by a discussion and conclusions."**

**Section 2.1**

**Line 111**: Rephrase.

**We rephrased this and the following sentence:**

**"Time versus position diagrams of measured and detided (residual) water levels are shown in Figure 2 a, b. The former shows the propagation of the tidal wave (the slope of contours gives the speed of propagation) along the English coast with water level amplitudes increasing southward towards the channel. The specific feature between stations Cromer and Vilssingen identifies the small amfidrome in front of the English Channel. This feature is not present in the detided data (Figure 2 b), the later resembles the atmospheric forcing (Figure 2 c, d). In the presence of large gradients in wind tendency (Figure 2c), the water surface tilts considerably compared to the case of small gradients (Figure 2b)."**

**Section 2.2**

**Line 134:** add references.

**We added the reference to CMEMS tide gauge product: CMEMS global ocean in situ near-real-time observations, https://doi.org/10.48670/moi-00036**

**Line 143:** avoid "very"**.**

**Lines 137-144 have been deleted and replaced by the following paragraph:**

**"Sea level data e.g. will form a 2D manifold. Different machine learning techniques, such as k-nearest neighbours algorithm, ensemble-based methods, and support vector machine (SVM) algorithms predict the posterior probabilities of a given dataset and are optimal for data compression. The different techniques are not equally well suited for detecting outliers, i.e. in deciding whether a given observation belongs to the same probability distribution. Outlier**

detection in high-dimension, or without any assumptions on the distribution of the inlying data is very challenging. SVM algorithms work well if  training data is not contaminated by outliers. Ensemble and k-nearest neighbour methods perform well for multi-modal data sets. Covariance estimators (in which category Principal Component Analysis, PCA, falls, too) degrade when the data is not unimodal. Autoassociative neural nets (AANN) combine the robust performance multi-modal data with the geometrical interpretability of PCA  to identify these situations…"

**Line 144:** Rephrase**. (Lines 137-144 have been deleted; see answer to Line 143)**

**Line 144:** check punctuation.  **(Lines 137-144 have been deleted; see answer to Line 143)**

**Line 155:** avoid repetitions: "Thus,…" **Done.**

**Line 192:** "presents" → "shows" **Done.**

**Line 204**: Is this choice based on a sensitivity analysis? The choice of the constraints/threshold errors should be explained in the text.

**The choice is based on results within the training period of AANN. We added on constraints/thresholds the following: "Thus, we postulate an ocean state as anomalous when the relative error exceeds 0.035 in the position of at least 3 gauges simultaneously for at least 3 hours."**

**Line 207-211:** "…, we trained the AANN based on the following data sets:

• sea-surface height outputs from CMEMS AMM15 (operational model) extracted at the tide-gauge positions (AANN_NEMO);

• observed total water levels from only 10 tide-gauge stations (with …) (AANN_less); and

• observed non-tidal water level residuals for the 14 gauge stations (AANN_resid). "

**We added information on data sets to Table 1:**

|  | training dataset | Spring (MAM) | Summer (JJA) | Autumn (SON) | Winter (DJF) |
|---|---|---|---|---|---|
| AANN_NEMO | NEMO model (14 data points) | 2 | 2 | 6 | 13 |
| AANN_less | tide gauge data (raw, 10 gauges) | 1 | 0 | 6 | 13 |
| AANN_resid | tide gauge data (de-tided, 14 gauges) | 1 | 1 | 6 | 13 |
| **AANN_ref** | **tide gauge data (raw, 14 gauges)** | **2** | **2** | **6** | **13** |

**Line 215:** "areal" ?

**We changed into "spatial"**

**Lines 225**: "…a comparison between observed, modelled and AANN emulated…".

**We show now rather the differences and thus reformulated: "A comparison of NEMO modelled and AANN emulated water level residuals with measured ones …"**

**Line 232:** "the reference model" ?. I assume this refers to AANN in Table 1 (See General Comments).

**Yes (see table 1 above), it is now referred to as 'AANN_ref' throughout the text.**

**Line 239:** "In the following Section,…": in the current version of the manuscript this Section is followed by the Conclusion. This part of the text should be reorganized.

**We removed 'section'.**

**Line 240:** It is unclear which ANN model error you are referring to (ANN_resid ?). The same applies tp NEMO model error: is it the error in NEMO ssh outputs?

**We clarified by referring to AANN names given in table 1 (see above) throughout the manuscript.**

**Comments about Figures**

**We increased resolution for all Figures to 400 dpi, made them more readable.**

**Figure 1:** The description in the caption should be improved as well as the labels in the figure (see wind tendency). **We splitted this Figure into 2 figures and changed Figure caption, too.**
**The captions now read:**
**"Figure 1: (a) Location of tide gauge stations used in this study and M2 cotidal chart. Additionally, snapshots of NEMO model water levels for two selected times are shown together with water level residuals at tide gauge locations (black lines) and a plane fitted linearly to these residuals (grey areas): (b) 7. January 2017, 16 UTC and (c) 11. January 2017, 20 UTC."**
**"Figure 2: Time versus position diagrams series of (a) measured water levels, (b) water level residuals, (c) wind tendency (defined as change in wind speed within one hour, here) and (d) pressure anomaly for January 2017 at tide gauge positions. "**

**Figure 3:** It is hard to read the labels.
**We have replotted the Figure with readable labels.**

**Figure 4:**
- Panels a, b, d, e (see previous comment on Figure 3).
**We have replotted the Figure with readable labels.**
- Panel c: It is very hard to read this panel, both time-series and labels. Also: show hours on x-axis.
**Fig.4c is now a separate figure showing a time versus position diagram of differences of NEMO model and tide gauge data, respectively. Labels are readable now.**

**Figure 5:** The panels should be displayed bigger as well as the labels in each of them.
**We splitted Figure 5 into two parts. Panels are bigger and readable now.**
Caption:
- "…a 14-day…" **Changed to actual periods: 1.-15. June 2017**
- "NEMO model error" – is it the AANN_NEMO ? Clarify. **Done. All reconstruction models are now shown in one Figure.**

**Figure 6:** Make sure the labels on top of the panels correspond with the description in the caption.
**Done. We have extended this Figure to contain all AANN reconstructions as well as those based on PCA. Additionally, a comparison with NEMO modelled results are shown.**

**Figure 7:** see comments on Figure 5.
**We have replotted as described for Figure 5.**

**Figure 8:** see comments on Figure 6.
**We have replotted as described for Figure 6.**

**Figure 9:** The panels can be displayed better. Avoid legend.
**We changed this Figure into a time versus position diagram.**

---

## Author Comment (AC2)

**Detecting anomalous sea-level states in North Sea tide gauge data using of auto associative Neural Network**

**Dear Agustín Sánchez-Arcilla, Dear Reviewer,**

**Thank you  motivatiforng us to improve further our paper, for reading it so carefully and for your valuable comments. We have corrected the  manuscript  accordingly and  provide the answers below.**

**Looking forward for any further instructions.**

**Best regards,**

**Kathrin Wahle**

**Review #2**

The paper presents an interesting application case of Neural Networks (NN) for the assessments of non-linear ocean dynamics, which can be used to identify and classify interesting non-linear ocean configurations and to study the ocean model's ability to reproduce them in detail. It might be helpful for the development of adequate parameterizations and for process implementations in numerical ocean models. This I think is an interesting point of the paper. The method could maybe be used to analyze the model's ability to represent events with anomalous correlations across modelled sea level stations, but this is not in the focus in the paper, which is dealing with the identification and classification of measured events. However, hydrodynamic model evaluation is part of the paper, so the evaluation of the model reconstruction error can be seen to be in the scope of the paper as well.

I suggest the authors to go a bit more in depth with the results analysis in chapter 3, to compare the two storm events with each other and maybe to analyze the model's ability to reproduce the anomalous correlation events of sea level time series (even if this is not in the focus of the study). I also suggest to elaborate the method part of the paper, to provide more information about the NN method and trainings method, which would be interesting for readers unfamiliar with NN methods, and to provide an analysis of the weaknesses of the method. I think that this can gain the paper more interest.

My background is more in operational modelling than in artificial intelligence. My statements with regards to the use of machine learning thus represent the perspective of an ocean modeler who wants to apply the method and not so much the perspective of a developer, who is more interested in the method itself. My remarks on the applied method should be seen in this context.

I recommend major revision, because I think that most of the figures have to be re-done and a part of the manuscript needs to be re-written. The statement does not refer to the general scientific quality of the manuscript, which I consider to be good.

**General comments:**

For me as a reader with no background in machine learning algorithms and IA, it would be helpful to understand the rationale for choosing the method to apply neural networks and if other, more recent methods could have been used as well. Neural Networks use "supervised learning" to train the algorithm. Other, more recent methods use "deep

learning" to train the algorithm. Could these be used as well? This could be of interest for readers from the operational modelling community, that are not necessarily experts in machine learning.

**Deep learning is based on artificial neural networks. Learning can be supervised, semi-supervised or unsupervised in either case. "Deep" learning refers to the use of multiple layers in the network. We use neural networks with one hidden layer with a non-polynomial activation function which were shown to be a universal classifier. Deep learning is a modern variation of this and can exhibit hundreds of millions of parameters. This circumstance hampers the interpretability of parameters as well as the explanation of results.**

Neural network, testing and training method: Which method was used for training the algorithm? In the manuscript, the method is introduced and the results are presented, but the information on how the method was trained and has been applied is rather limited. Have k-fold cross-validation methods been applied? How was the time series sampled? It would also be interesting to know how long it takes to train the algorithm. The analysis bases on the comparison of the reconstructions from different AAN-Networks that were trained using different data sets. So, the time it takes to train one AANN is an important factor in the efficiency of the analysis.

**We have extended the neural network part regarding choice of method, and training: "Different machine learning techniques, such as k-nearest neighbours algorithm, ensemble-based methods, and support vector machine (SVM) algorithms predict the posterior probabilities of a given dataset and are optimal for data compression. The different techniques are not equally well suited for detecting outliers, i.e. in deciding whether a given observation belongs to the same probability distribution. Outlier detection in high-dimension, or without any assumptions on the distribution of the inlying data is very challenging. SVM algorithms work well if training data is not contaminated by outliers. Ensemble and k-nearest neighbour methods perform well for multi-modal data sets. Covariance estimators (in which category Principal Component Analysis, PCA, falls, too) degrade when the data is not unimodal.**

**Autoassociative neural nets (AANN) combine the robust performance on multi-modal data with the geometrical interpretability of PCA to identify these situations."**

**This type of feedforward backpropagation network can be trained on a single CPU. Training time depends of the size of the dataset, but was several hours here.**

Influence of observation errors, especially under storm conditions. There is usually a lot of jitter in the observational data set, especially at the time of the sea level peak. Is there a need for quality control of the input data set? Which influence has the time resolution of the training and testing data set. Is hourly time resolution enough, to study the nonlinearities in the data set?

**The method is error tolerant against e.g. corrupt data of a single tide gauge. The method works well on raw (unprocessed) data. This is confirmed by the fact that AANN_ref (using raw data without any quality control) performs nearly equally well as AANN_resid (where data were processed using T-tide software package).**

**For a sea level to be detected as 'anomalous', AANN reconstruction error has to exceed a given error limit at three gauges for at least three hours. Thus, the method is not concerned with identification of small scale nonlinearities.**

Could the authors provide some analysis of the weaknesses of the used method, maybe in the discussion section. Is it possible to run the method for any configuration of observation stations? What would happen if the array of observation stations was extended eastwards into the Danish Straits and the Baltic Sea, which are less dominated by tides? Would the method still work?

**The weakness of the method, lies in the fact that one can not exclude the possibility of anomalous sea states not do be detected (false negative).**
**The method can easily be transferred to any other array of tide gauges. But, depending on the area, necessary amount of training data will vary.**

**General points:**

Use of English language: Please check the use of the english language and please re-write the text to improve the readability.
**We had the document checked by native speaker.**
Figures: The quality of the figures needs to be improved, which might also involve the restructuring of the figures. The resolution of the figures is fairly low. It is nearly impossible to read the text in the figures. Some figures have no title too, so that one has to guess what they are showing. Some of the sub-figures are not marked by letters.
**We have replotted all Figures with 400 dpi resolution. Additionally, we have restructured figures to improve figure content. All sub-figures are marked by letters. We have extended Figure captions.**
Nomenclature: Please check if the naming is consistent throughout the paper. An example are "anomalous spatial correlations", which sometimes are also called "unusual spatial correlations".
**We checked naming and now use 'anomalous' throughout the paper.**
My nomenclature throughout the review: In the text, "Line" with capital "L" refers to the line counter at the left side of the page, whereas "line" with small "l" refers to the line in the paragraph (line 6 in paragraph 2 of this chapter).

**0.Abstract**
**Line 8-15, page 1:** Abstract could be extended, to include all the topics covered by the study. The abstract in its current form serves mainly as a motivation. It is not describing the extend of the study, including the different application cases. I would also suggest to avoid specific, technical terminology or to re-phrase it, to not confuse readers with an operational background, which should be very interested in this paper. Terminology like "anomalous spatial correlations", "unusual spatial correlations", "lower dimensional subspace" should be introduced or re-phrased, to be better understandable.
**We have rewritten the abstract according to your suggestions:**
 **"Sea level in the North Sea is densely monitored by tide gauges. The data they provide can be used to solve different scientific and practical questions, among them validation of numerical models, as well as detection of extreme events. This study focuss on the detection of sea level states with anomalous spatial correlations by means of Auto Associative Neural Networks (AANNs), trained with different sets of observation- and model-based data. Such sea level configurations are related to nonlinear ocean dynamics, therefore neural networks appear to be the right candidate for their identification. The proposed network is able to accurately detect such anomalies and localize them. We demonstrate that the atmospheric conditions under which anomalous sea level states occur are characterized by high wind tendencies and pressure anomalies. The results show the potential of AANNs in detecting accurately the occurrence of such events. We show, that the method works with AANN trained on tide-gauge records as well as with AANN trained with model-based sea-surface height outputs. The latter can be used to enhance the representation of anomalous sea-level events in ocean models. Quantitative analysis of such states might help assess and improve numerical model quality in the future as well as provide new insights into the nonlinear processes involved. The method has the advantage of being easily applicable to any tide gauge array without preprocessing the data or acquiring any additional information."**

**1.Introduction**

**Lines 27—44: We have restructured this part of the introduction, separating now numerical models from satellite measurements and tide gauges.**

**Line 27, page1:** I would remove "largely". **Done.**

**Line 34, page2:** The sentence about operational model data does not fit here between satellite altimetry and tide gauge data sets. It should be moved further down.

**We extended the paragraph on recent developments in numerical modelling (Lines 37ff) and added the information there: "Recent important evolution in predicting sea-level in the North Sea was achieved in the framework of the development of the Northwest European Shelf forecasting system (e.g. O'Dea et al., 2012, Tonani et al., 2019) by enhancing model resolution to 1.5 km. Thus dynamical features such as coastal currents, fronts, and mesoscale eddies are better resolved, and improve model results, especially when compared to high spatial–temporal resolution observations."**

**Line 37-39, page 2:** Comment: When the model application for sea level observation data gap filing was mentioned, I thought first that this was what NEMO model results were used for in this study. It should maybe be mentioned at a later stage that this is not done.

**We now state: "However, tidal gauges do not provide information about the basin-wide patterns of sea level. Furthermore, some of these data are not continuous; different gauges do not always operate simultaneously, and there are gaps in many of the records. Therefore, a consistent dataset combining the gains of numerical model results with tide gauge measurements would be beneficial."**

**Line 47, page 2:** "Thermohaline forcing", is that coming from runoff of rivers?

**Yes. It comes from the sea surface as well as the lateral open boundaries.**

**Line 49, page 2:** What is a "perfect model"? Is this a model without prediction error? I would reformulate the sentence using a different word.

**We have used 'sophisticated' instead.**

**Line 58 (and below), page 2:** The term "anomalous sea-level states" or "anomalous situations" should be properly introduced. It is not a general term that is already defined elsewhere. In the manuscript, the term is used in two ways: (1.) as a term describing nonlinear conditions with strong forcing, i.e., storm conditions and (2.) as situations with significant reconstruction error of the AANN method. But are these two definitions really synonymous? Reconstruction errors might also originate from the choice of the trainings data set (non-anomalous situations that were not considered during the training, stations with non-significant correlations), observational errors (drift in the data set), unconsidered variables (dependency on waves, etc.) and maybe more. Can these errors be excluded from the analysis?

**You are right, the two terms are no synonyms. We clarified, that the term "anomalous" refers to reconstruction error and is not a synonym for non-linear conditions:**

**"From analysis of training period, we postulate an ocean state as anomalous when the error e exceeds 0.035 at at least 3 gauges simultaneously for at least 3 hours. "**

**Line 58-61, page 2:** Comment: The paragraph starts with saying what you want to use AANN's for and ends by saying that AANN's are good in reproducing the data set they were trained for. Logically this is not fitting. I would connect this paragraph with the next one.

**We reformulated this paragraph and moved it to the end of the introduction:**
**"In contrast to these applications, we will focus on the identification of situations in which spatial correlations between tide gauge measurements deviate greatly from the dominant principal ones. Usually, the entirety of tide gauge measurements in the North Sea will show specific spatial correlations (changing in time with the tide). However, in anomalous situations (e.g. localized**

**storms) these correlations may drastically change. We use autoassociative neural nets (AANNs, Kramer, 1992) to detect such sea-level states. Atmospheric conditions related to such situations might aid further understanding and future developments in sea-level prediction."**

Another point is, that the AANN method is at least able to reconstruct weakly non-linear events. So, a failure of the method means that the ocean state should be highly nonlinear.
Is it possible to provide a classification for the non-linearity of the observed model state? At which point is an ocean state "non-linear" and at which point is it becoming "anomalous"?

**There is a clear definition measured by the Rossby number (inertia/Coriolis). However, by using only gauge data we cannot make a diagnosis. What you maximum can do is to analyse the distributions. Non-gaussianity is an indication of non-linearity.**

**Line 69, page 3:** "NNs" instead of "NNss" **Done.**

**Line 80, page 3:** What are "mean correlations"? The term has not been introduced properly.

**You are right. We changed to 'principal' and explained: '… usually, the entirety of tide gauge measurements in the North Sea will show specific spatial correlations (changing in time with the tide). However, in anomalous situations (e.g. localized storms) these correlations may drastically change…'**

**2.Methods**

I would suggest to split the chapter into two parts: Data and Methods, or to rename the chapter to "Data and Methods". The chapter is dealing with the two aspects.

**We renamed this chapter 'Data and Methods' as suggested by you.**

**Figure1: page 4 (and all following figures):** Please improve the quality of the figure in terms of resolution. Maybe you need also to restructure the figure to make it better readable. I can not read the names of the sea level stations and the titles in sub-figures (be). In the captions, the letters indicating the sub-figures should be presented together with the parameters that are shown. "Black bars" are rather "black lines". How have the water level residuals been calculated (that should be mentioned in the text)? I could not find this information in the manuscript.

**We improved the quality of all Figures in terms of resolution and readability. Additionally, we rearranged the Figures. Here, we changed in caption 'bars' -> 'lines'. For de-tiding we used software package T-Tide and give reference to it in the text.**

**Line 85-105, page 3-4:** This paragraph belongs still to the introduction. It describes the model area and the general hydrodynamic of the North Sea, not the method that has been applied in the analysis. Instead, there should be an introduction into the use of the various data sets.

**We shifted this paragraph to introduction section.**

**Line 109-114, page 4-5:** It is unclear under which aspects the tide gauge observation stations were selected. Why were only southern and western North Sea tide gauge stations selected and not, for example, Danish stations?
More information about the observation data sets could be provided: data provider, monitoring period, frequency of the observations, data quality (amount of flagged data). Has the data been quality controlled? I know that at least some centers provide data to CMEMS without quality control. What about the time frequency of the data set? Is hourly data enough for the analysis, especially for the analysis of storm cases?

**Tide gauge data from CMEMS are hourly data (even though some stations measure at higher rates). We used the data as they are. We wanted to develop a simple but robust method without the necessity of data pre-processing. Stations were chosen according to completeness and availability of dataset. We added this to the text:**

**"Observational sea level data along the North Sea coast have been obtained from the historical and near real-time (NRT) dataset of the Copernicus Marine Environment Monitoring Services (CMEMS). The observations were taken from the respective in situ products for the NorthWest Shelf area product (Copernicus Marine In Situ TAC Data Management Team, 2020, http://marine.copernicus.eu/) with hourly resolution. The NRT in situ quality controlled observations are hourly updated and distributed within about 24–48 h after acquisition. From these, we have chosen 14 gauge stations according to completeness of data availability."**

**Line 115-133, page 5:** It is unclear at this point why NEMO model data sets play a role in the analysis. There should be a short introduction into the scope of the study, explaining the different data sources and their role in the assessment.

The introduction of the NEMO model configuration could be extended as well. I'm for example not entirely sure if the model run on its own or if it run as a component of the GCOAST model system, and which other components of GCOAST were included as well. Information about tidal forcing (boundaries, tidal potential), the initial data set and the model spin-up and forecast period should be required as well. Did the model use nesting in the transition area between the North Sea and the Baltic Sea. 3.5 km seems to be a rather coarse spatial, horizontal resolution for this area. Has the quality of the model with regards to sea level predictions been analyzed? If yes, could the authors provide a reference to the model validation study.

**We now motivate the usage of NEMO model in the introduction, by stressing:**

**"Therefore, a consistent dataset combining the gains of numerical model results with tide gauge measurements would be beneficial.-A similar exercise was undertaken recently by Madsen et al. (2019) for the Baltic Sea and by Zhang et al. (2020) for the North Sea. Zhang et al. use machine learning to reconstruct the sea-level variability in the North Sea using observations from 19 coastal tide gauges and data from numerical models. Noteworthy, they concluded that a relatively short-time record contains the most representative characteristics of sea level dynamics in the North Sea. While this was clear for the tides, it was not so obvious about changes in sea level caused by the atmosphere."**

**We have extended the description of the GCOAST model as follows:**

**"For the experiments discussed later in this study, we also use data from the GCOAST (Geesthacht Coupled cOAstal model SysTem) circulation, wave and ocean model (Madec, 2017, Staneva et al., 2017, 2021, Bonaduce et al., 2020). The wave-current interaction processes are momentum and energy sea state dependent fluxes, wave –induced mixing and Stokes-Coriolis forcing. The model area covers the Baltic Sea, the Danish Straits, the North Sea and part of the northeast Atlantic Ocean ) with 3.5 km horizontal resolution. The data used in the present study are only for the North Sea region shown in Figure 1. The ocean circulation model is based on the Nucleus for European Modelling of the Ocean (NEMO v3.6). The wave model is WAM (cycle 4.7), a third-generation wave model, which solves the action balance equation without any a priori restriction on the evolution of spectrum. The two models are two -way coupled via the OASIS3-MCT version 2.0 coupler (Valcke et al., 2013)..."**

**Regarding resolution, we agree that in the transition zone between Baltic and North Sea, 3.5 km resolution might be too coarse to resolve transport. We added the following information, too:**

**"Boundary conditions at the open boundaries (temperature, salinity, velocities and sea level) are taken from the AMM7 model (O'Dea et al., 2012) distributed by the Copernicus Marine Environment and Monitoring Service. They have a temporal resolution of one hour with 7 km horizontal resolution."**

**In Staneva et al. 2021 (see references) validation of GCOAST data with tide gauges can be found.**

**Line 130, page 5**: It is stated that ERA-5 atmospheric and wave parameter are provided by C3S. Have the wave parameters been used in this study? Otherwise, I would only

mention the atmospheric product.

**GCOAST results are from a coupled (NEMO/WAM) model run, boundary values for the wave model WAM are taken from C3S.**

**Line 134-145, page 5-6:** The autoasssociative neural network method is introduced (chapter 2.2). However, the more general description (first paragraph, Line 135 to 145) requires some further explanation and re-formulation:

The term "multivariate data" should be explained. I know the term only from data assimilation. I guess it is used here to point out that extensive, interdependent hydrographic data sets are measured that in the context of the AANN method are described as "multivariate data".

Another point is the definition of the sub-space of a given data space. What do you mean with "factors" (**line 3, first paragraph of section 2.2**)? Are you comparing the number of non-linear interaction processes between different monitoring parameters with the number of parameters? This is not so clear in the text. I think it needs further explanation or clarification. The way it is introduced here (at least the way I read it) is that the "sub-space" describes the reconstructed data set, not the original data set. So, errors in the reconstruction lead to situations when the original observed data sets are not an element of the reconstructed sub-space, in anomalous situations. Maybe this becomes clear after you have introduced the method, but its not so clear here.

**We have rewritten and extended this paragraph on AANN method. We avoid the term 'multivariate'. We clarified that we talk about 'data compression', when talking about 'subspace'. We introduce the method as:**

**'… Similar to PCA, this method aims at dimensionality reduction, but expanding the concept of orthogonal vectors to principal curves. The combination of such nonlinear components best describing the variability in the training data can be used to reconstruct the data themselves. …'**
**And further:**

**"Different machine learning techniques, such as k-nearest neighbours algorithm, ensemble-based methods, and support vector machine (SVM) algorithms predict the posterior probabilities of a given dataset and are optimal for data compression. The different techniques are not equally well suited for detecting outliers, i.e. in deciding whether a given observation belongs to the same probability distribution. Outlier detection in high-dimension, or without any assumptions on the distribution of the inlying data is very challenging. SVM algorithms work well if training data is not contaminated by outliers. Ensemble and k-nearest neighbour methods perform well for multi-modal data sets. Covariance estimators (in which category Principal Component Analysis, PCA, falls, too) degrade when the data is not unimodal.**

**Autoassociative neural nets (AANN) combine the robust performance of multi-modal data with the geometrical interpretability of PCA to identify these situations."**

**Near Line 140** (line 6, 1th paragraph of section 2.2) you speak about a "proxy". What do you mean exactly by this?

**We rephrased this paragraph (see above). It was not clear.**

**Figure 2 (b):** The way the fitting lines have been drawn in figure are a bit arbitrary. It is not so clear to me why not 6 neurons were chosen or 8 neurons. Is the method very sensitive to the number of bottleneck neurons?

**No, the method is not very sensitive, thus 6 or 8 neurons would do as well. It is the same compromise as with PCA: how much described variance is enough? How many leading modes one chooses is a bit arbitrary, too. Nevertheless, it is common practice to derive the number of bottleneck neurons as we did it.**

In **chapter 2.2** and sometimes in other parts of the document, the term "model" might be used in ambiguous ways. It usually refers to the "AANN model", but might also refer to

the "NEMO model". I would suggest to always write the full name of the model.
In the manuscript, the method is introduced and the results are presented, but general information on how the neural network has been trained is not provided (see general points at the beginning of the review).
**We gave names to all AANN's in table 1 and use those throughout the discussion:**

|  | training dataset | Spring (MAM) | Summer (JJA) | Autumn (SON) | Winter (DJF) |
|---|---|---|---|---|---|
| AANN_NEMO | NEMO model (14 data points) | 2 | 2 | 6 | 13 |
| AANN_less | tide gauge data (raw, 10 gauges) | 1 | 0 | 6 | 13 |
| AANN_resid | tide gauge data (de-tided, 14 gauges) | 1 | 1 | 6 | 13 |
| **AANN_ref** | **tide gauge data (raw, 14 gauges)** | **2** | **2** | **6** | **13** |

**Concerning training of AANN's, we added the following:**
**"During training, the AANN constructs a model, that captures the posterior probability distribution of a given data set.**
**At the beginning of the training, the outcome of the NN will differ largely from the target output. The mean squared relative error per neuron e is iteratively minimised during the training by backpropagating the error through the NN and adjusting free parameters according to a gradient descent scheme."**

**Anomalous sea-levels and their relationship with atmospheric conditions:**
This chapter falls into 2 parts, dealing with the analysis method (Line 200-235) and analysis of application cases (Line 240-280).
**3.1 Analysis method:**
The analysis method uses arbitrary limiters, which depend on the application at hand. The values for the threshold error, the minimum number of tide gauges and the minimum exceedance time of the threshold could be motivated with the AANN analysis results for the North Sea, either the entire testing year or the two application cases for 2 storms.
**The limiters are based on analysis of training period. We clarified in the text:**
**"From analysis of training period, we postulate an ocean state as anomalous when the error e exceeds 0.035 at at least 3 gauges simultaneously for at least 3 hours. "**
**Line 200-205, page 8:** I would use "sub-set of tide gauges" rather than "several" tide gauges. **We deleted "several".**
**Line 200-205, page 8**: What is the given threshold error? It is not mentioned in the manuscript.
**We added: "… error e exceeds 0.035" (see above).**
**Line 220-225, page 9:** What does "resemble" here mean? Are the graphs AANN_less and AANN_resid visually identical to AANN?
**The error distributions of AANN_less, AANN_resid and AANN_NEMO are visually very similar to the one of AANN_ref. Only, AANN_NEMO's tail of the distribution is shortened compared to the others. That is the reason to show it separately.**
**Line 220-225, page 9:** I would not say that the model physics is constrained linear processes, which follow a Gaussian statistic. Especially turbulence is non-linear. One can say maybe that the model processes are not highly non-linear and that non-linearities are

often dampened away by dissipation. It is also so, that interactions between processes can amplify nonlinearities in the model.

**Yes, you are right. We deleted that statement.**

**Line 220-225, page 9:** I don't understand the second sentence. What does the observational data errors have to do with AANN_NEMO specifically? Are they not the same for all reconstruction models?

**AANN_NEMO is trained on NEMO model data. All other AANN's are trained on observational data.**

**Line 220-225, page 9:** You refer to Fig. 4c, which I can not read well.

**We have replotted Figure 4c as a time versus position diagram of all tide gauges.**

**And we have rewritten the passage, you are referring to:**

**"The error distributions of AANN_less, AANN_resid and AANN_NEMO are similar to those of the AANN_ref. The tail of the AANN_NEMO error distribution is shortened, making the modelled data less suitable for novelty detection. However, a comparison of NEMO modelled with measured ones (Figure 4c) shows that anomalous situations (4., 12. and 14. January 2017) reflect themselves in indicative deviations between the two."**

**Figure 4 (c):** How have the water level residuals been calculated? Tidal Harmonic Analysis? How many components have been used?

**We used software package T-tide and refer to it in the text. We use 47 components.**

I can't see the figure well, due to its low resolution, but at some stations there seem to be still some harmonic signals. What about the reconstructed time series? Are the results of AANN_resid shown? Were the water level residuals of the NEMO model sea levels calculated? Maybe you are showing water level anomalies rather than residuals.

**We have replotted this Figure as time versus station diagram of errors between measured sea level (and residuals) and NEMO, respectively (see above).**

**Line 230-235, page 9:** It is not entirely clear which criterion was used, as the threshold error is undefined. What is the validation period that is analyzed in table 1?

**In the text, it reads now: "…an ocean state as anomalous when the error e exceeds 0.035 at at least 3 gauges simultaneously for at least 3 hours…"**

**The testing period is the year 2018. It is given in the caption of the table.**

**Table 1:** Which period does the analysis cover? How does the validation period relate to the training or testing period?

**In the table caption it is clarified that table 1 refers to testing period 2018. Training period are the years 2016/17.**

**3.2 Application cases:**

The analysis should be extended with a more detailed discussion of the results. Often the facts are presented without analyzing them in detail. The figures 5-8 seem to be rather separated from the discussion in the text. Sometimes a feature is singled out, like the sea level difference (observation-model) series at Lowestoft (fig. 5e), but all the other results in the figure 5e are not discussed and the results of the other sub-figures 5a-f are not discussed either. The same is true for figure 7 (Storm Burglind). The assessment is qualitative, not based on a statistical evaluation of model errors, especially water level peak errors. It seems also to be unclear, why the NEMO sea level prediction errors are discussed, as they do not seem to be directly related to the reconstruction error. For Storm Heinrich, for example, the reconstruction error is largest for stations in the southern North Sea (Dover, Oostende, Europlatform, Vlissingen), whereas the largest model error is occurring at Lowestoft. So, the conclusions are a bit uncertain.

I think the figures 5d and 6b,c,d and figure 7d and 8b,c,d for the two storm cases: Heinrich (June 2017) and Burglind (January 2018) are at the heart of the assessment. The analysis of the two cases shows different results in the reconstruction, which are briefly

discussed, but could be presented in a more consistent way. It would be rather interesting to analyze the ability of the hydrodynamic model to reproduce the observed anomalous correlation events figure 5d and 6d, as well as figure 7d and 8d.

One more comment: The exact dates for the two storm events should be given, as the names of storms vary from country to country.

**Thank you for this comment. We have extended the discussion for the two storm cases, e.g.: ('Heinrich')**

**"… As a result, the measured sea-level (Figure 8a) exceeded the AANN_ref modelled sea-level over several hours along the western Dutch coast and southeastern British coast (Figure 9 a). AANN_ref and AANN_NEMO (Figure 9b) errors are largest at Oostende, where the measured low tide is higher than expected by AANNs. Occurrence of the largest error shifts to Dover for AANN_resid and to Den Helder for PCA (Figure 9e, f). The NEMO model errors (Figure 9c) are largest for Vlissingen, where it under- overestimates the measured values for high and low tides, respectively. Consisteltly, AANN_less does not detect the event (Figure 9d), indicating its origin in anomalous spatial correlations among gauges around the small amphidrome. Thus, the forcing resulted in a localized (exceptional) increase in sea level at gauge station Den Helder. The snapshot in Figure 7c visualizes this finding at the time the AANN largest errors occurred, 10 hour delayed to wind tendency."**

**Line 240-241, page 10:** The reference to figure 5 seems to be incorrect. Are you sure that you don't want to refer to figure 6b and 6c?

**Yes, you are right. We apologize for the error. The sentence is referring to Figures 6b and c.**

**Line 240-255, page 10-11:** Storm Heinrich: Removing the southern North Sea stations Dover, Oostende, Europlatform, Vlissingen leads to a better reconstruction of the sea level (Fig. 6c). Does this mean that the non-linear events with (anomalous spatial correlations) come from these 4 stations?

**Yes.**

The results: measured sea level minus reconstructed sea level (AANN) (Fig 5d) demonstrate the reconstruction error (related to anomalous spatial correlations) and Fig. 6c seems to proof this. Figure 6b seems to show that the reconstruction error is not related to the tides, which likely means that it is driven by meteorological forcing. NEMO seems to be able to reproduce some of the nonlinearity features at Oostende and Vilssingen, but not at Dover. Why is that the case? PCA seems to be able to catch the event, but it also shows an event for Den Helder. Why is that the case? Which method is having a problem here?

**The NEMO model resolution is 3.5 km spatial resolution. The closest model point to Den Helder is too far away to include in the analysis. Thus it is not shown in the results.**

**Apart from that, principal components of PCA and "principal curves" of AANN are not the same. Thus, equality of results of the two different methods is not to be expected.**

**Line 255-280, page 11-13:** Storm Burglind: Removing tides seems to lead to a negative reconstruction error at the 4 Southern North Sea stations: Dover, Oostende, Vlissingen (Fig. 8b). What does this mean? Is the trained AANN_resid method reproducing nonlinearities that are not in the measured data set. Is this evaluated as a reconstruction error as well? In contrast, the AANN_resid method is generating a positive reconstruction error at Den Helder. Does this mean that atmospheric forcing is stronger in driving nonlinearities there? Is this related to the training method?

**AANN tries to reproduce its input. It does not know about non-linearities or meteorology, etc. From what AANN_resid has learned during training, residuals should be larger around Dover and smaller at Den Helder. This cannot easily be related to atmospheric forcing. It simply states, that this specific distribution of residuals in the study area was not met during training (or at least not as often as the output distribution).**

Removing the southern North Sea stations Dover, Oostende, Europlatform, Vlissingen leads to a reconstruction of the sea level (Fig. 6c) with some positive error. Does this mean that the non-linearities were stronger in the central North Sea? Can this be proven by training an AANN that does not consider other stations? It seems that the separation into station (AANN_less) suits better the case of storm Heinrich. NEMO seems to be able to reproduce some of the nonlinearity features, which should be mentioned, but they are not as pronounced as the observed ones. Differences could be analyzed in detail.

**This is line with the reference model AANN_ref. Stations around the first amphidrome don't seem to be affected but for all other stations measured sea level exceeds AANN reproduced ones. This is, what constitutes the 'anomalous' sea state here.**

**Line 270-280, page 12-13:** The reconstruction using PCA seems to work less well for Storm Burglind than it does for Storm Heinrich. Is this because only leading modes were considered? The number of PCA reconstruction modes is not mentioned in the manuscript. If this is the reason, then it should be made clearer in the text.

**PCA reconstruction (as well as the AANN's) works less well during a stormy winter period, than in summer. We used 7 leading modes (analogue to number of bottleneck neurons) and clarified in the text.**

**Line 272:** What is a false alarm?

**Here: when PCA reconstruction error is high, but AANN's reconstruction error is below threshold value.**